# Pharmaceutical Applications of Molecular Tweezers, Clefts and Clips

**DOI:** 10.3390/molecules24091803

**Published:** 2019-05-09

**Authors:** Amira Mbarek, Ghina Moussa, Jeanne Leblond Chain

**Affiliations:** 1Gene Delivery Laboratory, Faculty of pharmacy, Université de Montréal, H3C 3J7, Montréal, QC, Canada; a.mbarek_bm@yahoo.ca (A.M.); ghina.moussa@umontreal.ca (G.M.); 2Univ. Bordeaux, ARNA Laboratory, F-33016 Bordeaux, France; INSERM U1212, CNRS UMR 5320, ARNA Laboratory, F-33016 Bordeaux, France

**Keywords:** molecular tweezers, molecular switches, clefts, clips, responsive systems, drug delivery, biosensing, imaging, controlled release

## Abstract

Synthetic acyclic receptors, composed of two arms connected with a spacer enabling molecular recognition, have been intensively explored in host-guest chemistry in the past decades. They fall into the categories of molecular tweezers, clefts and clips, depending on the geometry allowing the recognition of various guests. The advances in synthesis and mechanistic studies have pushed them forward to pharmaceutical applications, such as neurodegenerative disorders, infectious diseases, cancer, cardiovascular disease, diabetes, etc. In this review, we provide a summary of the synthetic molecular tweezers, clefts and clips that have been reported for pharmaceutical applications. Their structures, mechanism of action as well as in vitro and in vivo results are described. Such receptors were found to selectively bind biological guests, namely, nucleic acids, sugars, amino acids and proteins enabling their use as biosensors or therapeutics. Particularly interesting are dynamic molecular tweezers which are capable of controlled motion in response to an external stimulus. They proved their utility as imaging agents or in the design of controlled release systems. Despite some issues, such as stability, cytotoxicity or biocompatibility that still need to be addressed, it is obvious that molecular tweezers, clefts and clips are promising candidates for several incurable diseases as therapeutic agents, diagnostic or delivery tools.

## 1. Introduction to Molecular Tweezers, Clefts and Clips

Host-guest chemistry has experienced tremendous development in the past decades. Initially inspired by Nature, synthetic receptors have been exploited to understand and mimic biological processes, such as molecular recognition, protein interactions [1] or DNA intercalation [2]. Once the biological process has been elucidated, synthetic receptors could be used to interfere with a pathological condition, leading to the development of drug candidates, diagnostic agents or drug delivery systems (DDS) [3]. For instance, several macrocyclic receptors such as the water-soluble calixarenes, cucurbiturils, and cyclodextrins have proven their utility as DDS [4]. Nevertheless, non-cyclic receptors, known as molecular clefts, clips or tweezers, are less explored in pharmaceutical applications. This review is aimed at collecting the pharmaceutical applications of molecular tweezers, clefts, and clips, using a comprehensive approach to relate some structural features to their pharmaceutical application.

The structures we are interested herein consist in open cavities with one or two binding “arms”, which are able to complex a guest through non-covalent interactions, such as hydrogen bonding, hydrophobic or van der Waals forces, aromatic stacking or metal coordination [5]. Several appellations have been used to designate such non-cyclic synthetic receptors, according to the authors who introduced them [6]. This short historical background presents these pioneering studies, their structures and related appellations.

The term “molecular tweezers” was introduced for the first time by Whitlock in 1978 to describe a caffeine-based receptor (**1**, Figure 1) [7]. According to Whitlock, molecular tweezers are a receptor in which two flat arms, generally aromatic, are separated by a more or less rigid spacer, and converge to provide a pocket for guest binding. The distance between the two aromatic side arms is close to 7Å, suitable for inclusion of aromatic guests through π-π interactions.

Molecular clefts were then described by Rebek in 1985 as rigid receptors exhibiting convergent functional groups directed toward each other and separated by a spacer [8]. The spatial disposition of the functional groups defines a space where a guest can fit [9]. In the first C-shape structure, an aromatic platform was connected to two cyclohexane moieties displaying convergent carboxylic acids (**3**, Figure 1). Conversely to Whitlock tweezers, rotation was hindered by steric effects and the pre-organized structure allowed the binding of amino acids or nucleic acids [10,11,12]. Molecular clefts can be tuned to accommodate guests of different dimensions or sizes with a high degree of selectivity.

Klärner, then, discriminated molecular tweezers from molecular clips according to their structure and guest binding mode [13]. In the structures Klärner developed, di- and trimethylene-bridged receptors with naphthalene or anthracene moieties form complexes by clipping flat molecules between their aromatic sidewalls, resulting in so-called molecular clips (**4**, Figure 1), whereas in tetramethylene-bridged receptors the side arms are expanded and compressed to accommodate the guest, according to the working principle of molecular tweezers (**2**, Figure 1).

Finally, the term “molecular switches” has been used by Feringa to describe synthetic structures, including molecular tweezers, where a motion is involved (**5**, Figure 1) [14]. Afterwards, this term was largely used to describe a system involving a conformation change [5].

Several reviews have described the structural diversity of molecular tweezers [5,6] or have focused on specific series [15,16,17]. Nevertheless, no review has screened the pharmaceutical applications of such promising structures, ranging from drug candidates or biosensors to drug delivery systems. This review aims at presenting a comprehensive molecular tweezers’ description to foster new applications of current or future synthetic receptors. Therefore, in this review, we have included the molecular tweezers, clefts, clips and switches that have demonstrated a proof-of-concept related to a pharmaceutical application. To facilitate the reading, we present first the use of tweezers for the recognition of biological guests, generally achieved by rigid structures and often translated into drug candidates or biosensing. In a second section, we illustrate how the dynamic and stimulus-sensitive switches have been explored in bioimaging or drug delivery systems.

## 2. Recognition of Biological Guests

### 2.1. Nucleic Acid Recognition

Nature has developed complex systems to recognize selectively biological guests, such as DNA or proteins. Interfering with this process enable a pharmaceutical control over gene expression and regulation. Indeed, many drugs have been developed to interact with DNA, either by intercalation (e.g., doxorubicin, mitoxantrone), groove binding (e.g., netropsin), covalent cross-linking (e.g., cisplatin), alkylation of bases (e.g., chlorambucil) or by inducing DNA cleavage (e.g., etoposide, bleomycins) [18]. All these mechanisms impair DNA replication and result in cytotoxic activity, mainly exploited in antibiotics or cancer treatment [18]. Similarly, molecular tweezers have been developed as a pharmaceutical tool to interact with DNA. Their selectivity has been exploited to recognize specific sequences of DNA, in order to detect and correct DNA defects.

#### 2.1.1. Purine Bases Recognition

Rebek et al. developed molecular clefts based on Kemp’s triacid connected to an aromatic platform, resulting in a tridimensional pocket able to bind nucleobases via π-stacking and base pairing interactions. For example, compound **6** (Figure 2) with a naphthalene moiety associates 9-ethyladenine from either Hoogsteen or Watson-Crick faces [19].

Various aromatic moieties were investigated to promote aromatic interactions with adenine. Extended aromatic platforms such as anthracene favored the Watson-Crick bonding, whereas addition of bulky groups to naphthalene platform was found to favor the Hoogsteen mode interactions due to steric effects [19]. Aside from π-staking, the number of hydrogen bonds (two or three) play an important role in nucleobase (adenine or guanine) recognition. For instance, adenine complexation within receptor **6** is stabilized stable due to the formation of bifurcated hydrogen bonds between the amine of the substrate and the carbonyls within the cleft. Diederich’s group further investigated the binding mode of such tweezers in both solution and solid state. Combination of crystallographic, NMR and computational studies revealed that adenine slightly prefers Hoogsteen over Watson-Crick base pairing, whatever the aromatic platform linked to the imide core [20]. Introduction of an acetylene linker between the imide and aromatic platform improved the spatial positioning of adenine in the cleft, improving the π-stacking between the naphthalene and the nucleobase rings (**7,**
Figure 2). This complex offers less hydrogen bonds than receptor **6**, the authors were able to isolate the contributions of imide H-bonding from π-stacking in the binding [21]. In a similar way, the cleft **8** (Figure 2) showed affinity towards guanosine derivatives by base-pairing [22]. This structure **8** was recently exploited by Roleira et al. to complex the antiviral acyclovir, a guanosine analog. Receptor **8** formed a 1:1 complex through three H-bonds and π-stacking, similarly to the complementarity observed in DNA guanine-cytosine base-pairing. This was one of the first reports of molecular tweezers as potential drug carriers [23].

Another type of adenine receptors has been developed by Plante et al., based on a pyridine bis-imidazole framework (**9**, Figure 2). In the presence of a metal ion, such as Cu^2+^, the molecule adopts a tweezers conformation, allowing for heteroaromatic guests complexation by metal and aromatic bonding [24]. Selectivity for adenine vs. other nucleobases was observed by UV-Vis.

#### 2.1.2. Trinucleotide Repeat Recognition in Myotonic Dystrophy Type 1 (DM1)

In parallel with Rebek’s studies, Zimmerman developed rigid molecular tweezers featuring a dibenzacridine spacer holding a pair of parallel acridines (Figure 3) [25]. The *syn* orientation of the planar acridines and their 7 Å distance separation promotes aromatic guest binding through π-sandwich complexation. Introduction of a carboxylic acid in the cleft provided additional H-bonding possibilities and promoted complexation of adenine derivatives in non-polar organic solvents [26] (**10,**
Figure 3). This complexation did not occur with an ester instead of a carboxylic acid substituent, showing thereby the crucial role of the hydrogen bond in the binding process. Unfortunately, these tweezers tend to self-associate into dimers in aqueous media, resulting in poor water solubility and limited biological applications. Therefore, Zimmerman moved to asymmetrical tweezers using a triaminotriazine, a thymine recognition unit, connected to an acridine intercalator (**11**, Figure 3). This structure exhibited high affinity for T-T mismatches involved in expanded CTG repeat causing Myotonic Dystrophy type 1 (DM1) [27]. This affinity was maintained with CUG repeats of RNA and the complex inhibited the sequestration of the alternative splicing factor muscleblind-like 1 (MBNL1) by the aberrant CUG repeat, restoring the active function of the protein MBNL1 (Figure 3) [27]. However, molecule **11** was highly cytotoxic and showed both poor solubility and cell permeability in a cellular model of DM1. Addition of an oligoamine side chain improved the solubility and cell penetration but showed moderate inhibitory effect on MBNL1-CUG complex [28]. Dimerization of the tweezers through optimized polyamine linkers allowed to generate a multivalent ligand with improved affinity for CUG repeats. In particular, dimer **12** achieved good water solubility, cellular and nuclear permeability and efficient dispersion of (CUG)_n_/MBNL1 complexes in a DM1 cell model [29]. The acridine intercalator was then removed from the structure for toxicity reasons and the tweezers’ structure was shortened to two triaminotriazine recognition units linked by a bisamidinium linker (**13** and **14**, Figure 3). Unlike previous tweezers, these receptors exhibited an unstacked conformation, which provided flexibility to bind three consecutive CUG units in the major groove with high selectivity. This binding reversed the toxicity induced by (CUG) repeats and corrected the missplicing of mRNA in a DM1 *Drosophila* model in vivo [30]. This study with receptor **13** was the first report of an in vivo application of nucleotide-targeted molecular tweezers. The structure was then dimerized using click chemistry to improve the affinity, which allowed to reach a nanomolar inhibition constant [31]. Recently, the tweezer was conjugated to a tris(2-aminoethyl)amine moiety (**14**, Figure 3) which provided the ability to cleave the DNA in addition to completely inhibit nuclear foci formation in a cellular model of DM1 [32]. Subsequent in vivo experiments showed that **14** was able to improve two separate disease phenotypes in a DM1 *Drosophila* model namely, adult external eye degeneration and larval crawling defect [32]. Taken together, these findings show that Zimmerman’s bisamidinium ligands are promising lead compounds for DM1 treatment, through the inhibition of MBNL1 binding to (CUG) repeats.

#### 2.1.3. Mismatch Recognition

Aiming at specific interactions with nucleobases, Nakatani et al. explored the 2-acylo-1,8-naphthyridine moiety which exhibited complementary H-bonding profile with guanine. When two units are connected through a linker (**15**, Figure 4), the dimer is able to detect G-G mismatches [33], in solution or bound to a SPR surface, creating a biosensor [34]. 

In this recognition process, each unit is bound to a guanine base through hydrogen bonding and π-stacking (**15**, Figure 4), and the linker structure plays a critical role in the binding efficiency [33,38]. Indeed, slight changes in the linker structure and removal of one carbonyl group (using 2-amino-naphtyridine, **16** and **17**, Figure 4) resulted in selective recognition of C-C mismatches with high selectivity, regardless of the flanking sequences [39]. Structure investigation lead to the optimization of specific tweezers for A-A mismatches, using *N*-methoxycarbonyl-2-amino-1,8-naphthyridine (MNCD, **18**, Figure 4) and a very short linker [35]. These compounds, referred as “molecular glue” by the authors, were subsequently used to bind to DNA defects, and reverse DNA hybridation upon heating (thermal degradation [40] or UV irradiation, **19,**
Figure 4 [36]). Finally, conjugating a diamino-naphthyridine (DANP, **20,**
Figure 4) to a pyrene moiety yielded a fluorescent sensor of cytosine bulges (**15,**
Figure 4) [37]. Indeed, pyrene fluorescence is quenched by DANP, but recovers its fluorescence when DANP recognizes and complexes a cytosine bulge in DNA [37]. Expanding the aromatic ring to a tricyclic unit allowed a better recognition of cytosine in a C-C mismatch or a CCG trinucleotide repeat [41].

#### 2.1.4. Trinucleotide repeats in Fragile X Mental Retardation Syndrome and Huntington disease

Nakatani’s specific recognition units were also used to detect unusual expansion of trinucleotide repeats, observed in several neurodegenerative diseases, similarly to Zimmerman’s approach. For instance, naphthyridine carbamate dimer (NCD, **21,**
Figure 5) developed by Hagihara et al., was able to intercalate into CGG repeats, a trinucleotide repeat found in Fragile X Mental Retardation Syndromes. The NCD binds to guanine bases and triggers the flip of the widowed cytosine out of the DNA helix (**21**, Figure 5) [42]. This complexation stopped the Taq DNA polymerase replication activity and resulted in an improved detection of the genetic defect. Coupling to a ferrocene moiety resulted in an electrochemical probe, facilitating the diagnostic of the disease [43]. Asymmetrical tweezers enabled the selective recognition of trinucleotide (CAG) repeat, an expansion mutation causing Huntington’s disease [44]. In this case, the 2-acylo-1,8-naphthyridine still binds guanine whereas azaquinolone on the other side of the tweezers recognizes adenosine [44].

More recently, NCD dimers were crosslinked by disulfide bridges or covalent linkers, bringing four hydrogen bond partners in the close vicinity of the four guanines bases from a CGG/CGG mismatch (**22,**
Figure 5) [45]. The complex was able to form a hairpin structure on RNA, which stopped the translation process in vitro [46]. RNA-Seq revealed that the tetramer significantly impacted gene expression in HeLa cells, suggesting that genes can be regulated by synthetic ligands (**23,**
Figure 5) [47].

#### 2.1.5. DNA Binding and Anticancer Agents

Several dimers of 1,8-naphthalimide have been designed from the pioneer work of Brana et al. (Table 1) [48]. These compounds bind to DNA, inducing damages leading to cytotoxicity and showed interesting anticancer efficiency in lung, colon, melanoma cancer models. Variations of their structures to improve specificity and anticancer activity has been recently reviewed [49]. Although these structures usually do not show a stacked cleft conformation, some linker variations increased the rigidity of the structure, which became very similar to molecular tweezers (Table 1) [50]. However, few studies have demonstrated the specificity and selectivity of these structures for DNA sequences, which is a core property of molecular tweezers. Yet, these studies reveal the pharmaceutical potential of tweezers-like structures as anticancer agents.

Overall, by designing specific ligands to nucleobases, molecular tweezers were able to detect mismatches, bulges (i.e., unpaired bases) and trinucleotide repeats, involved in neurodegenerative diseases. They could be used as diagnostic tools, biosensors or anticancer agents as well as drug carriers. Further applications are expected in the next years since these ligands were found to interfere with DNA hybridation [36], replication [42], gene expression [47] and RNA translation [46].

### 2.2. Amino Acids and Protein Recognition

Molecular tweezers are biomimetic receptors inspired from the lock-and-key model of enzyme-substrate interaction. Therefore, they are excellent candidates for protein interaction, leading to pharmaceutical alteration of biological processes.

Rebek’s initial tweezers displayed affinity for adenine (**3**, Figure 1). Interestingly, switching for an acridine platform enabled the complexation of β-aryl amino acids: phenylalanine, tryptophan and tyrosine *O*-methyl ether [11]. Such amino-acids were bound by two receptors via hydrogen bonding and π-stacking, enabling their extraction from an organic liquid membrane.

Klärner and Schrader developed a library of tweezers and clips carrying norbornadiene and aromatic rings. These receptors provide highly electron-rich clefts able to surround various electron-poor cationic or neutral substrates via π stacking and π-CH interactions [13]. Among this library, belt-type tweezers (**2**, Figure 1) comprising alternating four methylene bridges and five benzene rings obtained via repetitive Diels-Alder cycloaddition were able to complex aliphatic guests such as lysine and arginine, whereas the other receptors show a preference for aromatic substrates. Compared to tweezers, the clip receptors (example **4**, Figure 1) with extended aromatic sidewalls (naphthalene or anthracene) and reduced methylene bridges display an open topography allowing the inclusion of sterically large guests into its cavity. Several applications have been explored by Bitan and collaborators using Klärner’s series of clips and tweezers, which are presented below.

#### 2.2.1. Enzyme Inhibition

The water-solubility of Klärner’s receptors as well as their binding properties can be tuned by grafting functional groups onto the central spacer. Interestingly, anionic phosphates promoted ammonium guest binding in water via additional hydrophobic interactions and coulombic interactions. Hydrophobic effect was found to play a critical role in the complexation process in water as self-association competed with guest inclusion depending on the size and shape of the tweezers. Particularly interesting are receptors **24** and **25** that overcome dimerization to form stable complexes with guests.

Phosphate clips (**24,**
Figure 6) have been shown to accommodate aromatic cofactors of enzymes, such as nicotinamide adenine dinucleotide NAD^+^, Thiamine diphosphate TPP and S-adenosylmethionine SAM [51]. Phosphate tweezers (**25,**
Figure 6) on their part have been shown to display specific recognition towards lysine and arginine, but do not bind to other amino-acids or cationic cofactors [52]. Both phosphate clip **24** and tweezers **25** (Figure 6) were found to inhibit the enzymatic activity of alcohol dehydrogenase ADH, an enzyme carrying multiple lysine residues on its surface and using NAD^+^ cofactor for its activity. Interestingly, the clip and the tweezers displayed different inhibition mechanisms (Figure 6). Indeed, due to its affinity toward NAD^+^, the clip **24** competes with cofactor binding, prevents NAD^+^ to associate to the parent enzyme and lead to cofactor depletion. Whereas the tweezers **25** selectively associate to the multiple lysine residues found in the enzyme, especially near the active site and inhibited the substrate binding (Figure 6). Similarly, phosphate clip **24** inhibited the enzymatic activity of glucose-6-phosphate-dehydrogenase (G6PD). The clip sequestrates NADP^+^ cofactor and occupies both the cofactor and the binding sites, involving several mechanisms of inhibition [53].

Very recently, inhibition of poly(ADP-ribose)polymerase 1, (PARP-1), a key enzyme for DNA quality control, was demonstrated with the same compounds **24** and **25**. Interestingly, the tweezer **25** was shown to bind lysine residues of the enzyme as well as the metal cation of the zinc finger, likely through its phosphate group. This specific positioning in the active site displaces DNA from the zinc finger and inhibits the enzyme activity in a non-competitive mechanism [56].

#### 2.2.2. Prevention of Protein Aggregation In Amyloidosis

Because lysine plays a key role in aberrant protein self-assemblies that cause various amyloid-related diseases, phosphate tweezer **25**, also known as CLR01, revealed as a potent inhibitor of amyloid aggregation and toxicity [16]. CLR01 selectively binds to lysine and arginine residues of amyloid proteins through a “threading mechanism” in which the amino-acid side chain is included within the cavity of the tweezers via hydrophobic interactions and the anionic phosphate groups connects to the ammonium of lysine or the guanidinium of arginine via columbic interactions (**25,**
Figure 6) [16]. Consequently, the molecular tweezers compete with the hydrophobic and electrostatic interactions involved in protein misfolding, inhibiting thereby the aggregation process (Figure 6, right). 

CLR01′s inhibitory effect was first screened on various disease-associated amyloidogenic peptides or proteins, including amyloid β (Aβ), protein Tau, islet amyloid polypeptide (IAPP), transthyrein (TTR), insulin, CT and β_2_-microglobulin (β_2_-m) [55]. CLR01 was able to inhibit the initial formation of β-sheets and subsequent fibrils of all these proteins, at a concentration correlated to the number, location and relative abundance of lysine in the protein sequence (Figure 6, right). CLR01 significantly prevented the toxicity induced by these proteins on prostate cancer cells. Interestingly, no or minimal activity was observed with the negative control CLR03, i.e., the same receptor lacking aromatic sidewalls, highlighting the role of sidewalls in lysine binding and amyloid fibrils inhibition (Figure 6, right) [55]. This study was a crucial discovery for molecular tweezers, showing that CLR01 was a wide-range aggregation inhibitor, while exhibiting very low toxicity. Further studies were focused on specific diseases to understand further the mechanism of action of CLR01 and evaluate its clinical potential (Table 2).

The most advanced results concern Alzeihmer’s disease (AD). Amyloid-β (Aβ) and Tau proteins are two major proteins involved in AD. By binding specifically to lysine and arginine of monomeric Aβ, CLR01 was shown to form small nanometric complexes, which structure precludes the formation of toxic oligomers. Interestingly, CLR01 was also able to dissolve pre-existing fibrils in a dose-dependent manner [16]. On neuronal culture and brain slices, CLR01 protected the neuron from the synaptotoxicity induced by Aβ, restoring the electrophysiological activity of neurons [57]. In a mice model of AD, molecular tweezers were shown to cross the blood–brain barrier, disassembled the existing Aβ and Tau aggregates and also inhibited synaptotoxic effects of oligomeric Aβ. The results were confirmed in rats, where the treatment with CLR01 (0.3 mg/kg) led to 52% reduction of the plaque burden, showing the efficiency was not limited to one model or species. Together with a thorough safety assessment of this drug candidate [58], these results raise good hope that CLR01 will process to clinical trials in the coming years.

CLR01 also was found to inhibit the toxicity of α-synuclein, a protein involved in the development of Parkinson disease (PD) and other synucleinopathies. The negatively charged CLR01 binds to Lys residues at the N-terminus of α-synuclein reversing the positive into negative charge. This phenomenon was found to disrupt molecular interactions inside the peptide chain, increasing its reconfiguration rate and therefore inhibiting oligomerization and aggregation processes. CLR01 not only interacts with α-synuclein monomers, but also induced disaggregation of preformed fibrils in vitro, similarly to Aβ protein [59]. In vivo, addition of CLR01 to the water rescued the phenotype and survival of a zebrafish model of α-synuclein toxicity [60]. In a recent study, intracranial injection of CLR01 on mice overexpressing α-synuclein reduced the protein level in the striatum and significantly improved the motor dysfunction of the animals. The moderate effect observed after subcutaneous injection suggests that the blood-brain barrier permeability should be further improved before clinical translation [61].

Similarly, CLR01 was found to reduce transthyretin (TTR) toxic aggregates involved in transthyretin amyloidosis. Here again, CLR01 triggered the formation of innocuous structures that do not proceed to amyloid fibrils. In vivo, subcutaneous infusion of CLR01 on a transgenic mouse model of Familial Amyloid Polyneuropathy (FAP) restored the TTR plasma level to healthy conditions, significantly reduced TTR extracellular accumulation in several organs and rescued tissue damage [62]. Although a 10-times higher concentration of CLR01 was needed to achieve the similar inhibitory effect of EGCG, a tea catechin reported as potent inhibitor of TTR, CLR01 is viewed as a promising lead for FAP.

Inhibition of Islet Amyloid Polypeptide Assembly (IAPP), involved in type-2 diabetes (T2D), followed a different pathway. Substoichiometric concentrations of CLR01 inhibited the fibril elongation, but more than 100 equivalents of CLR01 were required to completely dissolve the toxic oligomers, and inhibit IAPP’s toxicity [63].

In tumor resistance, some mutations of p53 protein result in p53 denaturation, aggregation and toxicity. Here again, CLR01 acted as an aggregation modulator, yielding intermediate-sized p53 aggregates with reduced cytotoxicity [64].

CLR01 was found to reduce cardiac protein aggregation caused by cardiomyocyte-specific expression of mutated αB-crystallin CryAB^R120G^. Robbins’ group showed that CLR01 inhibited CryAB^R120G^ aggregation and cytotoxicity efficiently in both cultured cardiomyocytes and mice models by improving proteasomal activity [65].

CLR01 was found to reduce the aggregation of mutant super-oxide dismutase involved in the motor-degeneration of amyotrophic lateral sclerosis (ALS). Unfortunately, if CLR01 treatment reduced the misfolding in the mutant protein in the spinal cord of mice, this was not translated into motion improvement for the animals [66].

Although CLR01 was mainly reported to bind arginine and lysine residues, the molecular tweezers were found to inhibit polyglutamine (polyQ)-related amyloid aggregation involved in Huntington’s disease [67]. Surprisingly, the tweezers exhibited affinity for the glutamate residues of the 17-residue N-terminal fragment (N17) of the huntingtin protein. This binding triggered structural rearrangements, modified the amphipathic nature of N17-domain and reduced aggregation in a dose-dependent manner. Interestingly, Zimmerman’s bisamidine-based tweezers also exhibited a neuroprotective role by reducing polyQ protein aggregation and suppressing neurodegeneration in vivo in a *Drosophila* model of polyQ disease [68].

#### 2.2.3. Modulation of Protein-Protein Interaction (PPI)

Interestingly, CLR01 can also modulate the interaction between two different proteins. In a major paper led by Ottmann’s group, crystal structures showed that CLR01 binds preferably to the Lys214. The selectivity of CLR01 toward lysine originates from threading the butylene chain of Lys through its cleft, allowing hydrophobic interactions with the tweezer arms, and coulombic attraction between anionic phosphate of the tweezer and cationic ammonium group of lysine. Moreover, complexation of lysine depends on the steric accessibility. Indeed, CLR01 was found to bind only the sterically favorable exposed Lys residues found in intrinsically unfolded proteins. For instance, among the 17 lysine residues of 14-3-3 protein, only five were likely to be complexed by CLR01 with a large preference for the exposed Lys214 over the others 4 lysine residues. Since Lys214 is positioned in the close vicinity of its central channel, CLR01 interfered with 14-3-3 protein-protein interaction, inhibiting for example the binding of the phosphorylated C-Raf and the unphosphorylated Exoenzyme S partner proteins in a dose-dependent manner [1]. Conversely, the same group showed in a subsequent study that CLR01 could stabilize the protein-protein interaction between 14-3-3 protein and Cdc25C binding partner, as much as 20-fold [69]. These findings originated from the bifacial recognition displayed by CLR01 toward 14-3-3 protein inside its cavity on the one hand and toward a basic residue of Cdc25C phosphatase via its exterior apolar aromatic recognition site, on the other hand. Such modulation of protein-protein interactions is regarded as promising therapeutic approaches to control cellular processes, in particular in cancer [69].

#### 2.2.4. Inhibition of Enveloped Viruses

More recently, antiviral applications of Klärner’s tweezers CLR01 have been envisaged. Münch et al. showed that CLR01 affected the formation of amyloid fibrils found in semen, the main vector of HIV transmission and antagonized viral infection [70]. In addition to modulate the aggregation mechanism, the authors demonstrated that the tweezers interacted selectively with lipid-rafts regions of the virus envelope, resulting in viral membrane disruption. This new mechanism was confirmed for other enveloped viruses including human cytomegalovirus, herpes simplex virus type 2, and hepatitis C virus, but the tweezers were ineffective against the non-enveloped counterparts [70]. CLR01 was also found to prevent Ebola and Zika infection in a dose-dependent manner, suggesting a potential as a broad-spectrum protective antiviral agent [71].

Overall, Klärner, Schrader, Bitan and co-workers have pioneered the field of biological application of molecular tweezers. The tweezers CLR01 is the most advanced molecular tweezers as a drug candidate. In vitro and in vivo studies have revealed its therapeutic potential for degenerative disorders like Alzheimer and Parkinson’s diseases. Drug development and clinical translation has been initiated by the characterization of drug-like properties, toxicology and safety profiles. In parallel, mechanistic studies have given significant insight on its binding to exposed lysine residues, thereby broadening the range of pharmaceutical applications to protein-protein interactions and antiviral agents.

### 2.3. Sugar Recognition

Designing selective hosts for small molecules is much more difficult than for proteins due to the limited possibilities of interactions. Nevertheless, several small molecules could present a biological interest and have been investigated as targets for molecular tweezers. In particular, carbohydrates, which are involved into many biological processes, can be used as biomarkers in several diseases (e.g., diabetes and cancer). Sugar recognition by synthetic ligands has involved supramolecular receptors; most of them present polydentate configuration to optimize hydrogen bonding display. The diversity of artificial sugar receptors has been recently reviewed [72]. Among them, few bidentate receptors can be identified as molecular clefts or tweezers.

Thiourea motives have been introduced in a bidentate structure to favor multipoint recognition and binding of glucopyranoside through hydrogen bonding (**26**, Figure 7) [73]. This multipoint hydrogen bonding, as well as the flexibility of the structure, promoted versatility of the accommodated guests, such as carboxylic acids and sugars.

More usually, boronic acids are exploited for sugar recognition, thanks to the covalent bonds they can form with sugar hydroxyl groups. James et al. acknowledge that the spatial disposition of the boronic acid moieties determines which saccharide is bound preferentially [74]. Their straightforward structure displays two boronic acids linked by an anthracene electron-rich group (**27**, Figure 7), and allows selective detection of glucose through Photoinduced Electron Transfer (PET), preferentially to other sugars. Their tweezers were able to detect physiological levels of glucose in the presence of other sugars [75].

Recently, Wang et al. have developed a symmetrical bis-boronic acid that mimics lectins and selectively recognizes sialyl Lewis X (sLe^X^) thanks to its cleft conformation (**28,**
Figure 7). Optimization of the linker structure, through peptide [76] or click chemistry [77], allowed to detect selectively cancer cells overexpressing sLe^X^ in vitro. Conjugated to BODIPY fluorophore, a single intravenous injection lighted up a subcutaneous xenograft in mice with high selectivity (**28,**
Figure 7) [78]. Since such carbohydrates are a common biomarker of many cancer types (e.g., liver, colon, breast, lung), such boronolectins display a great potential as targeted imaging agents for cancer diagnosis.

## 3. Stimuli-Responsive Tweezers for Drug Delivery

In addition to their affinity and selectivity, several molecular tweezers exhibit a dynamic behavior controlled by external stimuli such as pH, light, redox, temperature or metal [5]. Such stimuli induce a conformational switch, usually reversible, which is able to release the guest bound in the cleft and/or destabilize the self-assembly properties of supramolecular systems. Incorporated into nanomaterials, such as liposomes, this dynamic behavior can trigger the release of a drug specifically at the target site. In this section, we review significant examples of dynamic conformational switches used for drug delivery or imaging, triggered by the coordination of metals, protons or cations. It has to be noted that light switches constitute a large field of research in itself and are not included in this review. A recent review of their structure and applications has been recently reported [79]. Nevertheless, fluorescent probes involve more and more small molecules, which structures are close to acyclic receptors [80].

### 3.1. Metal Switches for Sensing or Delivery

Conformational change of a molecule upon metal complexation/decomplexation allows the construction of devices with both dynamic and luminescent properties, enabling chemical sensing. Terpyridine, widely used to build supramolecular complexes metals ions, was explored to produce U-shaped dynamic tweezers with a well-defined cavity for selective recognition and reversible molecular binding. Their photophysical properties are mainly due to a charge transfer between a transition metal and terpyridine unit (cation-π interactions). The earliest molecular terpyridine based switches able to reversibly interconvert between a U and a W shape upon metal coordination are derived from Lehn’s work (**29, 30**, Figure 8) [81]. In a first series, terpyridine spacer carrying either acridine or anthracene as recognition arms adopts an initial open W shape (**29,**
Figure 8). Complexation of Zn^2+^ cation brings the recognition arms in close proximity, resulting in a closed U shape, suitable for aromatic guest intercalation. When pyridine-pyrimidyne-pyridine chelating spacer was used instead of terpyridine, 3-state U/S/W conformational metallo-switches were generated. In this second series, the initially uncoordinated tweezers adopt a U-shape (**30**, Figure 8) favored by transoid conformation of pyridine-pyrimidine cycles, ideal for the binding of electron-donor aromatic guests. The coordination of a Cu(I) cation orients the heterocycles into their cisoid conformation resulting in an intermediate S-shape followed by an open W-shape, which triggered the release of the aromatic guest [81]. While pyrimidine-based tweezers **30** show an adequate U-conformation to form complexes with planar aromatic guests via donor-acceptor interactions, guest inclusion by W-initially shaped terpyridine tweezers **29** was only possible when the guest participates in zinc coordination. Upon guest binding, Lehn’s complexes display color changes arising from charge-transfer interactions between the chromophores arms of the tweezer and the guest. Although the system was not tested with a pharmaceutical compound, the binding and release of an electron-donor aromatic guest was controlled by zinc and copper cations, which are essential oligo elements [82].

Cycloplatinated complexes have also been used to produce conformational switches with luminescent properties. In this regard, Vives and co-workers utilized Pt(II)-salphen complexes with terpyridine spacer to design dynamic tweezers [83]. Metal coordination (Hg^2+^, Zn^2+^, Pb^2+^) to the central terpyridine folds the two Pt-salphen arms towards the binding pocket resulting in a U shape (**31**, Figure 8).

These tweezers can be reopened by addition of tris(2-aminoethyl)amine as competitive ligand, enabling the reversible process upon demetallation of the central terpyridine. In the open W shape, phosphorescent Pt-Salphen moieties exhibit structured emission with a high quantum yield and long lifetime. The emission properties are slightly modified in response to Zn^2+^ or Pb^2+^, but strongly quenched upon the selective intercalation of a second Hg^2+^. Such metallo-switches can be applied in the design of luminescent sensors based on phosphorescence enhancement or quenching upon cation recognition. While the closed form was supposed to accommodate a substrate between the two Pt-salphen arms, no complexation was observed upon addition of aromatics. These results were explained by the steric hindrance of the existing *tert*-butyl groups of the tweezer, preventing π-stacking interactions. Although adding an alkyne spacer to *tert*-butyl groups and reducing their number did not resolve the problem [86], changing the substitution position enabled efficient coronene complexation within the cavity (**32**, Figure 8) [87]. The former structure was used for nickel coordination, and resulted in molecular tweezers responding to three stimuli: metal coordination, redox reaction and guest binding. In this latter case, pyrazine guest was bound either upon nickel coordination or oxidation of the tweezers [85].

Another example of Pt-salphen-based luminescent molecular switch for guest binding was reported by Wang et al. [84]. When two competitive guests: o-nitrobenzyl dimethyl caged naphtol derivative and Pt(II)-2,6-diphenylpyridine complex were added to a bis-alkynylplatinum(II) terpyridine molecular tweezers, a stronger binding affinity towards Pt(II)-diphenylpyridine was promoted by Pt-Pt interactions and π-stacking (**33**, Figure 8). When exposed to UV light, the protecting group is cleaved, resulting in a greater affinity between the tweezer and the decaging naphtol derivative, favored by intramolecular hydrogen bonding. Subsequent decomplexation of cyclo Pt(II) 2,6-diphenylpyridine complex led to optical changes owing to suppression of Pt-Pt interactions. Such a 3-component system capable of controlled guest binding and release with distinct optical changes may serve as potential bifunctional supramolecular system for drug delivery and sensing.

### 3.2. Lipid Switches for Controlled Drug Release

Lipid nanocarriers such as vesicles, liposomes or nanoparticles are being increasingly used as smart drug delivery systems due to their biocompatibility, non-immunogenicity and high loading capacities. Stimuli-responsive lipid switches have been incorporated into the bilayer where they undergo a conformational switch, in response to the binding of ions, triggering membrane destabilization and cargo release [88]. It has to be noted that during the writing of this review, a concept review has been published on lipid switches, illustrating the originality of such systems [88]. Therefore, we will focus solely on lipid switches which obviously demonstrated a pharmaceutical application.

One of the most explored stimuli is pH. pH variations can be found in the gastro-intestinal tract (pH 2–8), in inflamed tissues (pH 6.5–7), in interstitial tumor tissue (pH 6.5–7) or within intracellular compartments, such as endosomes (pH 5–6) or lysosomes (pH 4–5) [89]. Although numerous pH-sensitive lipids rely on chemical hydrolysis of pH-sensitive bonds, fewer systems are based on a molecular switch. Guo’s group developed a library of ‘fliposomes’ and demonstrated their use as lipid helpers in liposomes formulations [90]. The key component of these lipids is a protonable trans-2-aminocyclohexanol (TACH) head group bearing two alkyne tails (**34**, Figure 9). Protonation of amine results in a ring ‘flip’ to a second cyclohexane chair conformation favored by intramolecular H-bond between amine and hydroxyl group, in which the lipid tails switch to axial position [91]. Structure-activity relationship studies of a library of flipids with various head groups and lipophilic tails demonstrated that (i) pH-sensitivity as well as encapsulation/release capacities are highly impacted by the nature of head group and the tail length [92]; (ii) the molar ratio of lipids in fliposomes can be tuned to balance between their stability and pH-sensitivity [93]. Fliposomes were able to induce significant gene transfection in vitro, although the results cannot be explained by the sole conformation switch of the flipids [94].

Further pH-sensitive lipids based on an anisole-pyridine-anisole tricycle were developed in our group (**35,**
Figure 9). Previous studies have demonstrated the pH-sensitive behavior of this triad when it was flanked with aromatic arms, allowing the selective binding and release of an anticancer agent [95]. Flanked with aliphatic chains, the tweezers are organized in a U conformation at physiological pH and insert into a lipid bilayer to form liposomes, stable for several months [96]. The protonation of the central pyridine favors H-bonds between nitrogen and hydroxymethyl groups, locking the lipid in a W-shaped conformation prone for membrane disruption and thereby drug release. The evaluation of a small library of lipid switches carrying different tail length and various heads groups revealed that lipids with tertiary amines displayed a pK_a_ close to 5, suitable for endosomal escape. The corresponding liposomes massively released their content upon acidification, reaching 88% release when pH dropped from 7.4 to 5 in less than 15 min [96]. Interestingly, a control lipid unable to switch conformation was included in the study and clearly demonstrated the importance of the pH drop to trigger the release. In vitro studies demonstrated intracellular delivery of hydrophilic compounds thanks to endosomal escape, without showing any toxicity signs. A second generation of lipids was designed for gene delivery, bearing a cationic headgroup to associate nucleic acids. Such cationic switchable lipids allowed efficient gene silencing in vitro on a hypercholesterolemia model, and were able to reduce in vivo by half the secretion of a liver protein after a single intravenous injection in mice [97]. In a recent work, switchable liposomes were used to carry both anticancer drugs and miRNA, improving the treatment of retinoblastoma in a rat model and in primary human cells [98].

Using calcium binding to trigger a conformational switch, Best et al. reported a calcium sensor, designed to undergo membrane disruption upon calcium binding [99]. The molecular switch features two long alkyne chains bound to the Indo-1 Calcium sensor. The latter has been widely used as intracellular calcium dye. Ca^2+^ coordination by carboxylic acids induced a spatial reorganization, moving the lipid chains away from each other (**36**, Figure 9) [99]. Membrane disruption and drug release capacity were demonstrated using fluorescent hydrophobic and hydrophilic model drugs. Although bivalent cations such as Cu^2+^ or Ni^2+^ triggered some release, Ca^2+^ yielded the greatest release rate, showing the selectivity of the sensor. These calcium-responsive liposomes provide an interesting approach for selective drug delivery since calcium is found in greater abundance in certain pathological states, such as cancer or malaria.

Interestingly, the chair-boat transition was exploited by other groups to trigger a lipid switch within membranes. Veremeeva et al. designed a bispidinone derivative grafted with two alkyl chains able to incorporate into liposomes at 25 mol%. Upon addition of CuSO_4_, the liposomes released their fluorescent cargo within one hour [100]. In a similar way, Takeuchi et al. used both stimuli Zn^2+^ and H^+^ to monitor their assemblies. Particles were formed upon addition of Zn^2+^, chelated by two amine groups of the pyranose cycle and acidification strongly modified their structure. Unfortunately, these structures were unable to interact with living cells [101].

Smith et al. have developed molecular probes based on zinc(II)-dipicolylamine (ZnDPA) (**37**, Figure 9). This small molecule is composed of two zinc cations chelated by pyridines and shows a strong affinity for anionic phospholipids, such as phosphatidylserine (PS). Therefore, they bind to negatively-charged membranes in vitro and in vivo, which can reveal disease or dead cells. Grafted with diverse fluorescent reporters, several ZnDPAs have been used and commercialized to image apoptotic cells in vitro, tumor necrosis in vivo, to monitor bacterial infection and cancer treatment [102]. In addition, the ZnDPA structure was optimized to bind and destabilize anionic liposomes [103], serving as a trigger for the release of 5-aminolevulinic acid (5-ALA), a precursor to the photosentizer protoporphyrin IX [104]. In this concept, anionic liposomes containing 5-ALA are first administered and the trigger is added to monitor the release of the drug on demand. Mechanistic studies have shown that these tweezers act as a membrane permeation agent, rather than a fusion or switch mechanism [105].

## 4. Summary and Outlook

Starting from simple synthetic receptors, supramolecular systems have grown in complexity and control leading to molecular motors and nanomachines, recently rewarded by the Nobel Price of Chemistry awarded to Stoddard, Sauvage and Feringa [106]. Such major breakthroughs are currently translated into applications, ranging from chemical processes (catalysis, purification) to biological investigation (biosensing, biological probes) and pharmaceutical applications (drug candidates, drug delivery systems) [107]. Molecular tweezers, clips or clefts, as defined in this review, belong to this supramolecular field and undergo the same process. They are feeding the pharmaceutical sciences with new materials, with controlled design and smart properties, for the detection, delivery and treatment of diseases. Since such tweezers are inspired from nature and the “key-and-lock” enzyme model, these synthetic receptors appear as natural and excellent candidates for pharmaceutical applications.

Recognition between the tweezers/clips/clefts and a substrate requires a complementarity between the receptors and their guests. In the examples described above, it is clear that the geometry or conformation of the substrates, the size of the binding cavity as well as the nature and the number of interactions determine the binding process. For instance, the dynamic metallo-switches described above require metal coordination to switch the host’s conformation to a U shape, favorable for guest inclusion. In the Klarner’s library, the geometry of the receptors and the size of their cavities were tuned by the number and the nature of arenes. Reducing the number of methylene bridges resulted in larger cavities able to surround large guests. The distance between both sidewalls can be tuned using benzene, naphthalene or anthracene naphthalene-spaced clips. Belt-type tweezers 2 that showed a better affinity toward aliphatic guests rather than aromatic ones, were found to undergo a substantial distortion of the receptor geometry upon complexation of benzene rings [108]. Finally, functionalization of the central aromatic unit with phosphonate or phosphate groups offers additional hydrophobic and Coulomb interactions for guest attraction into aqueous media. It is important to note that an advantage of C- or U-shaped receptors is their structural flexibility allowing the rearrangement of their open cavities in order to fit the size and topology of the guest [108].

We truly believe that molecular tweezers can be further explored in pharmaceutical applications, and gain to be known from the pharmaceutical sciences community. Therefore, in this outlook, we summarize the progress and envisage the future applications of tweezers in the following questions:

### 4.1. How to Use Molecular Tweezers?

To summarize the pharmaceutical applications of the molecular tweezers presented in this review, we could say that the tweezers are currently used as:(i)A new drug candidate. In this category, CLR01 is the most advanced compound, showing multiple applications and preclinical data. To progress in the drug development process, the drug-like properties need to be established (absorption, distribution, pharmacokinetics, metabolism, elimination) and might represent a challenge due to the generally hydrophobic aromatic structure.(ii)A biosensor, for the detection of various molecules or supramolecular assemblies, from cations to DNA mismatches. In this field, the highest challenge is the selectivity and sensitivity of the system in biological medium.(iii)A drug delivery system, usually with responsive properties. The conformational change offers a faster kinetics and a better controlled assembly over other responsive systems, such as hydrolysable polymers. Nevertheless, the biocompatibility and stability of such new excipients with biological media need to be investigated.

### 4.2. In Which Disease Could Molecular Tweezers be Used?

Such concepts have been explored in a large range of diseases, depending on the substrate or the biological process involved, as illustrated on Figure 10. Noteworthy, the diseases targeted by molecular tweezers match the current major health challenges and World Health Organization priorities: neurodegenerative disorders (such as Parkinson and Alzheimer’s diseases), cardiovascular diseases (e.g., Diabetes-2), cancer, and infections (HIV, malaria, Zika, Ebola, etc.). One can note that genetic disorders and orphan diseases are also a major focus for molecular tweezers. A new treatment for such diseases could benefit from the fast-track approval of regulatory agencies, accelerating the clinical trials and commercialization of innovative drugs.

### 4.3. What are the Next Steps for Pharmaceutical Development of Molecular Tweezers?

The past years have focused the efforts on optimizing the structure, to gain in selectivity and affinity for their biological targets. Such feature is definitely an asset for drug development, since it ensures a good efficiency and would limit the off-target or side effects. It has to be noted that the synthesis of such structures is already a challenge in itself because the 3-D structure is so precise that it might not suffer a variation of one or few atoms. The next challenges would concern their successful translation in vivo. While Klarner’s molecular tweezer CLR01 proved to be potent in the treatment of various amyloidogenic diseases and showed to be safe in animals [58], many synthetic receptors active in living cells failed to achieve the same efficacy in vivo [66]. Indeed, the ability of the drug to reach their target in vivo rely on their biopharmaceutical properties, also called drug-like properties. Determined early in the drug development process, such properties help the selection of promising drug candidates, which would display favorable absorption properties, as well as distribution, metabolism and excretion, among others. Few studies have focused on these parameters, except for CLR01, which describes its pharmacokinetics and brain toxicity profile [58]. Noteworthy, the temperature might also be a critical parameter, since the in vitro binding studies are usually carried out at room temperature and might not represent what is occurring in living animals.

Another—and related—issue would concern the toxicity of such compounds. Although inspired from nature, their polycyclic and/or aromatic composition is far from the natural materials of living organisms (e.g., amino acids and oligonucleotides). The poor water solubility and the aromatic structure of several tweezers might promote the toxicity of such compounds. Unfortunately, Zimmerman’s tweezers showed poor water solubility and significant toxicity in vivo experiments, making them inappropriate for therapeutic applications [15]. Here again, the toxicity should be addressed in vitro and in vivo according to standard procedures for drug development, in particular DNA damages and teratogenicity for DNA binding agents.

### 4.4. Towards New Molecular Tweezers?

To overcome biocompatibility issues, synthetic mimics of biomolecules made of nucleic acids, aptamers, nucleotides and peptides have been developed for targeted therapies. For instance, DNA switches are extensively used as sensors [109] and nucleic-acid or amino acid-based aptamers are used to bind selectively biological targets [110]. However, synthetic mimics of dynamic molecular tweezers or switches are still very rare and are not exhaustively studied. One exciting example is the synthetic G-protein coupled receptor developed by Clayden and co-workers [111]. They designed a switchable membrane-bounded synthetic foldamer carrying a ligand binding pocket and two interconversional pyrenes in both sides. The researchers demonstrated, using pyrene fluorescence, that upon chiral ligand binding the receptor undergo a conformational switch that is transferred through the phospholipid bilayer [111]. This activity, similar to G-protein-coupled receptors toward ligand binding has been demonstrated in artificial bilayers and in living cells.

Finally, another promising development of molecular tweezers is their use as non-covalent linkers in supramolecular structures. Wang and coworkers exploited their host/guest properties to monitor the assembly of chain-like polymers, dendrimers and incorporated them into nanoparticles [112]. As compared to polymer and dendrimer materials, they allow the dissociation of the supramolecular assembly into small molecular weight molecules, easier to excrete in the kidneys. Lots of creativity is seen in this field, which holds great promise and versatility for the controlled design of drug delivery systems.

## Figures and Tables

**Figure 1 molecules-24-01803-f001:**
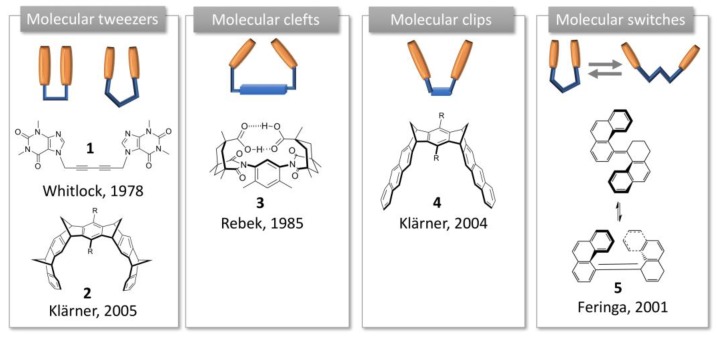
Schematic representation of the molecular tweezers, clefts, clips and switches introduced to describe acyclic synthetic receptors, and their first reported examples.

**Figure 2 molecules-24-01803-f002:**
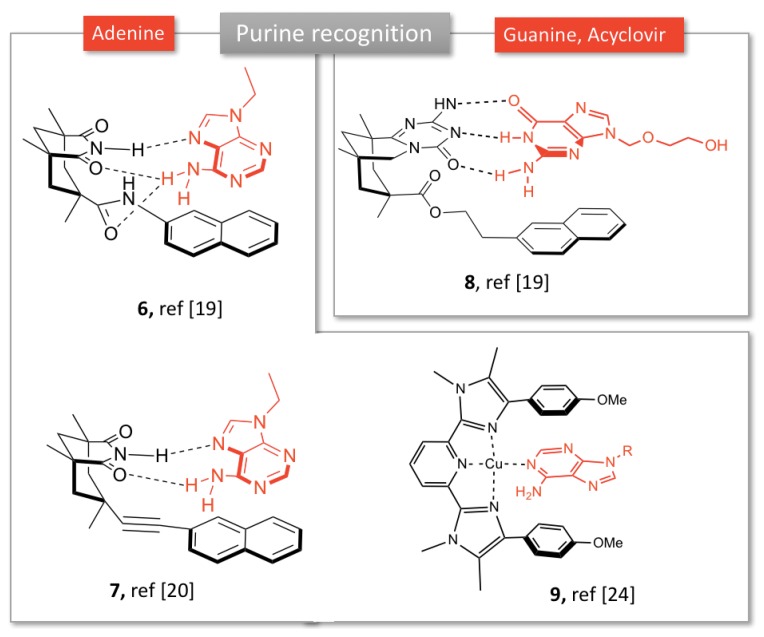
Examples of synthetic receptors for adenine and guanosine recognition.

**Figure 3 molecules-24-01803-f003:**
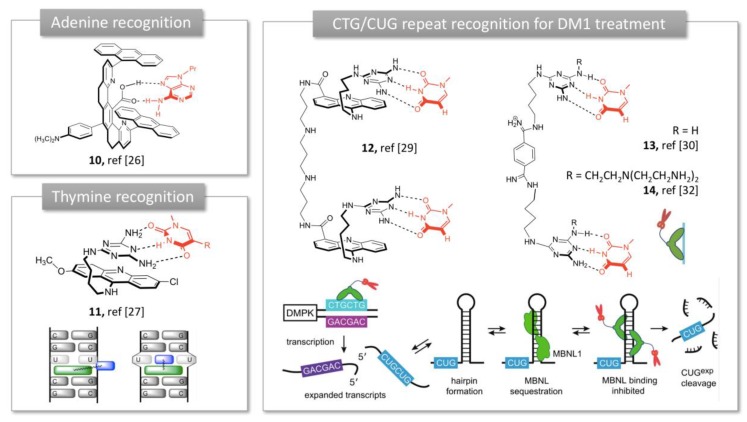
Receptors developed by Zimmerman: from molecular tweezers to triaminotriazine ligands for trinucleotide repeat recognition and cleavage. Adapted from [15].

**Figure 4 molecules-24-01803-f004:**
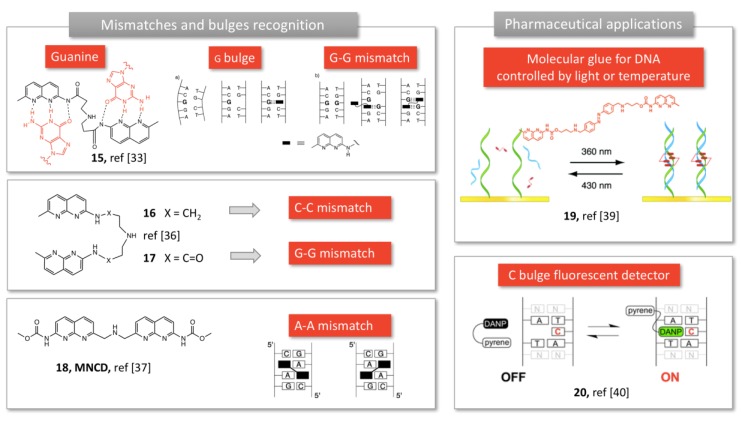
Molecular tweezers developed by Nakatani et al. to detect bulges and mismatches in DNA. Adapted from [33,35,36,37].

**Figure 5 molecules-24-01803-f005:**
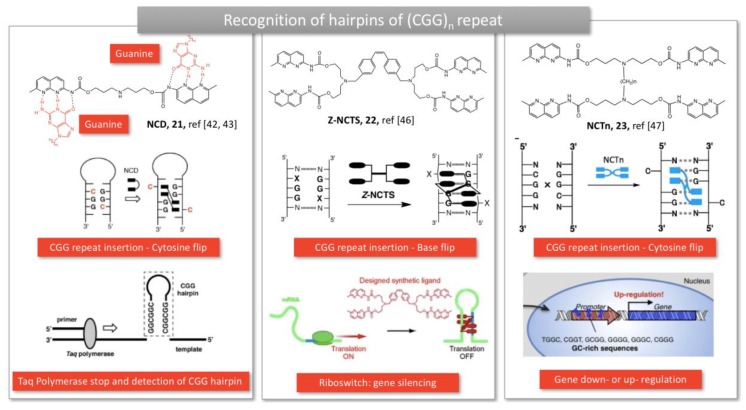
Molecular tweezers developed to detect CGG repeat causing hairpins and their pharmaceutical applications. Adapted from [42,46,47].

**Figure 6 molecules-24-01803-f006:**
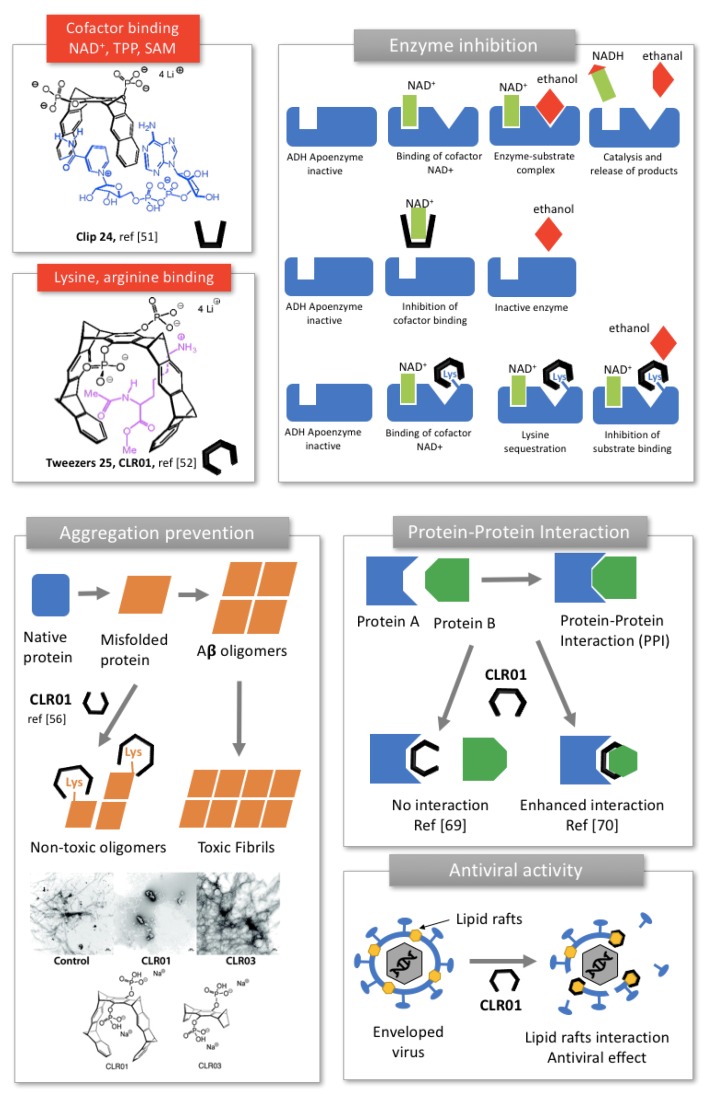
Phosphate molecular clip and tweezers developed by Klärner and their applications as enzyme inhibitors, illustrated by the alcohol dehydrogenase (ADH), protein aggregation inhibitors illustrated by amyloid β (Aβ) fibril aggregation, protein-protein interaction modulator and antiviral agent. Adapted from [54,55].

**Figure 7 molecules-24-01803-f007:**
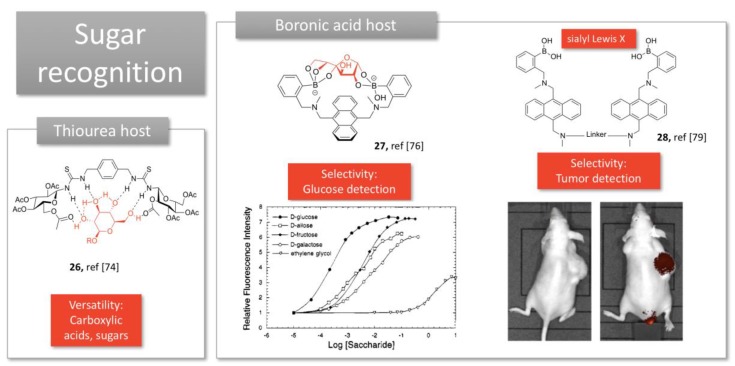
Examples of molecular tweezers used for sugar detection. Adapted from [75,78].

**Figure 8 molecules-24-01803-f008:**
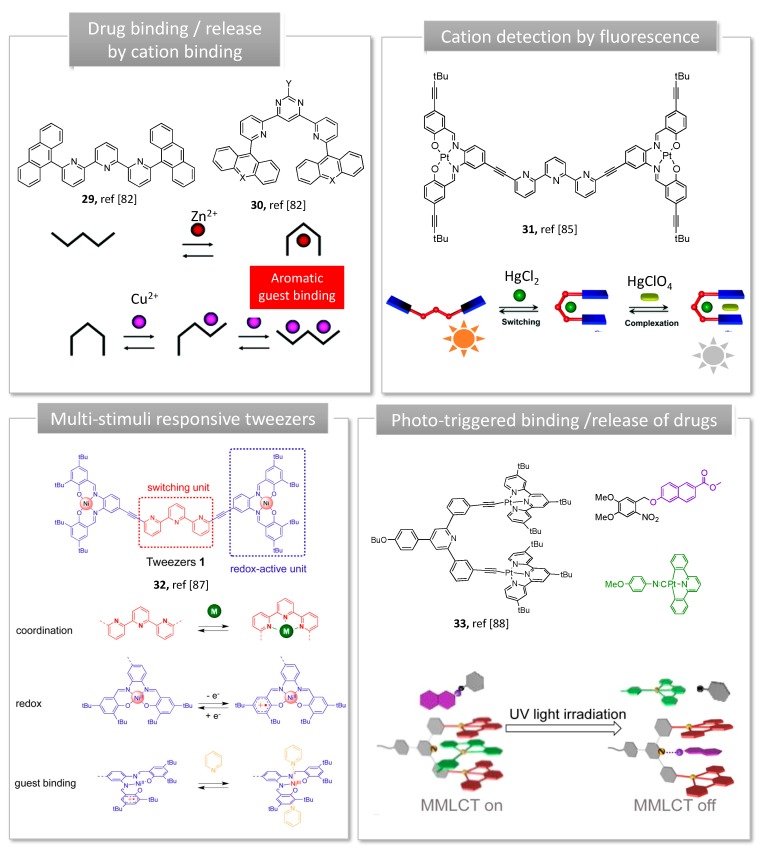
Examples of metal-switching tweezers. Adapted from [81,84,85,86].

**Figure 9 molecules-24-01803-f009:**
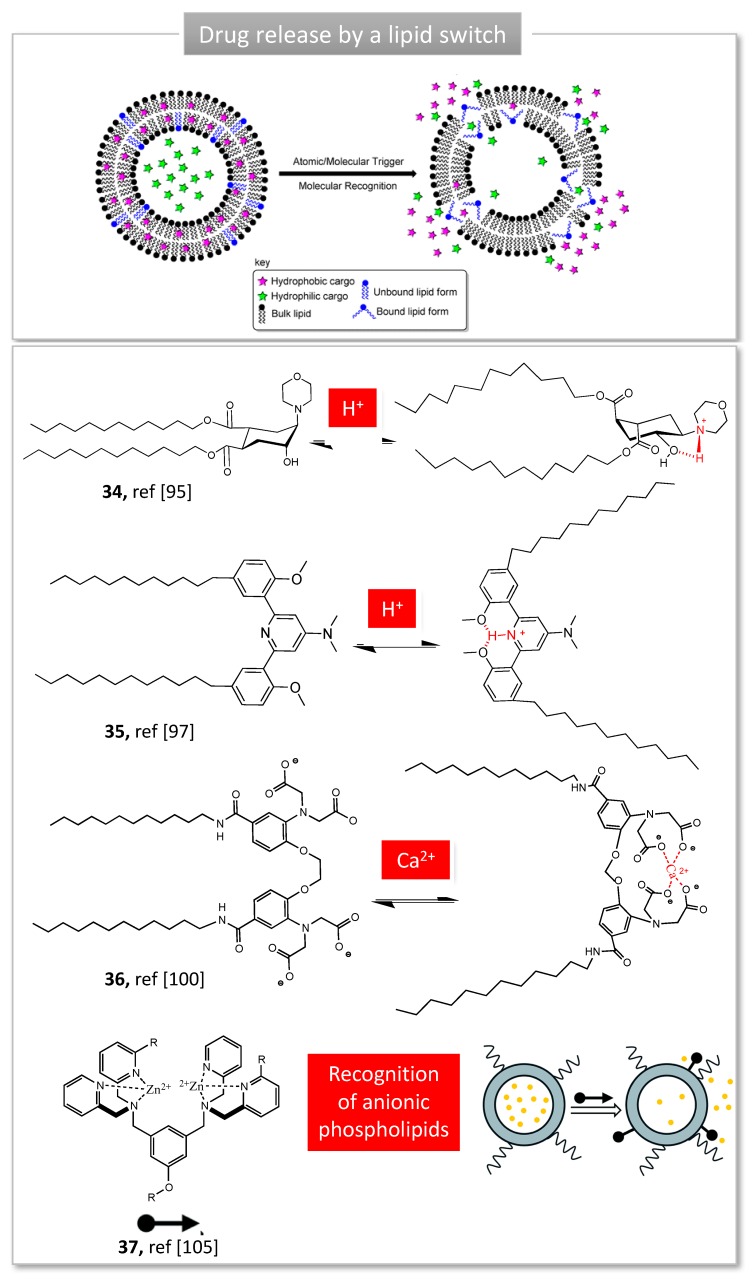
Examples of lipid switches used for stimulus-responsive drug release. Adapted from [88,104].

**Figure 10 molecules-24-01803-f010:**
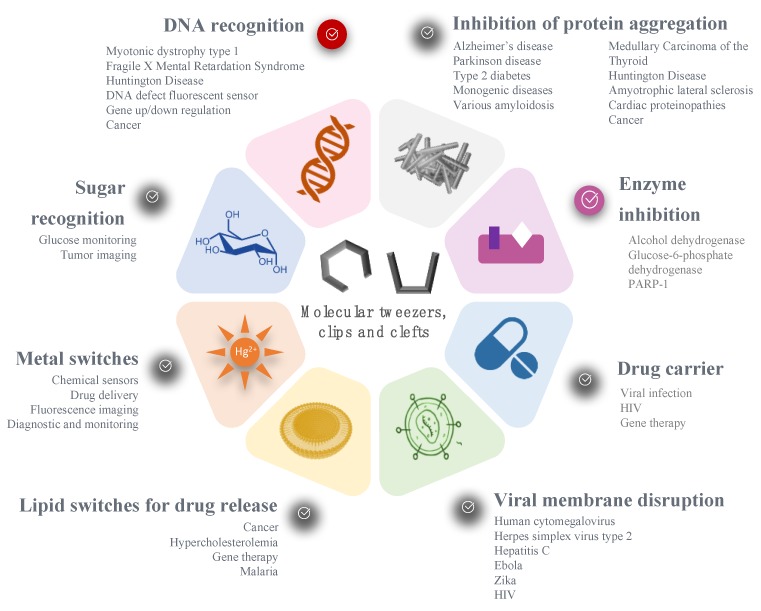
Overview of the pharmaceutical applications of molecular tweezers, clefts and clips.

**Table 1 molecules-24-01803-t001:** Examples of molecular tweezers based on 1,8-naphthalimide with anticancer properties. Adapted from ref [49]. ND: not determined.

Molecular Tweezers	Anticancer Activity	Cancer Model	Ref.
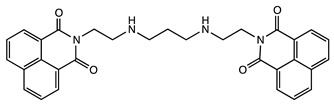	Bisintercalator in the hexameric d(ATGCAT)_2_ sequence	Human colon cancer HT29	[48]
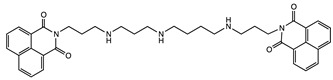	DNA fragmentation	Colon adenocarcinoma Caco-2 and HT29	[49]
	Chromatin condensation Caspase activation		
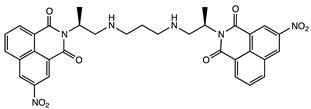	Interference with T and U incorporation	Leukemia	[49]
	DNA breaks	Solid tumor models	
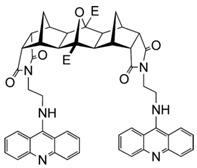	DNA unwinding	ND	[50]

**Table 2 molecules-24-01803-t002:** Amyloidogenic proteins inhibited by CLR01 and their associated diseases.

Protein	Associated disease	Refs.
Amyloid β-protein	Alzheimer’s disease	[55,57]
Tau	Alzheimer’s disease, Parkinson disease	[55,57]
α-synuclein	Parkinson Disease, synucleinopathies	[59,60]
Islet amyloid polypeptide (IAPP)	Type 2 diabetes	[55,63]
Transthyretin (TTR)	Familial Amyloid Polyneuropathy	[55,62]
14-3-3 adapter protein	Cancer, bacterial infections	[1,69]
Insulin	Injection-related nodular amyloidosis	[55]
β_2_-macroglobulin	Dialysis-related amyloidosis	[55]
Calcitonin (CT)	Medullary Carcinoma of the Thyroid	[55]
Polyglutamine core of HTT exon 1	Huntington Disease	[67,68]
p53	Cancer	[64]
Superoxide dismutase	Amyotrophic lateral sclerosis	[66]
Amyloid fibrils in semen	HIV	[70]
CryAB^R120G^	Cardiac proteinopathies	[65]
Viral envelope	Ebola, Zika	[70,71]

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
