# Peer review of "Pharmaceutical Applications of Molecular Tweezers, Clefts and Clips"

_molecules, 2019, doi:10.3390/molecules24091803_

Round 1

Reviewer 1 Report

In this review, the authors summarized the synthetic molecular tweezers, clefts and clips for pharmaceutical applications. The receptors were found to selectively bind biological guests, including nucleic acids, sugars, amino acids and proteins enabling their use as biosensors or therapeutics. Moreover, they could be used as dynamic molecular tweezers for the controlled motion in response to an external stimulus, and also be utilized as imaging agents or in the design of controlled release systems. It was proved that molecular tweezers, clefts and clips could be promising candidates for several incurable diseases as therapeutic agents, diagnostic or delivery tools. The manuscript can be published in Molecules.

Fig. x and Figure x in the text should be unified.

Some figures should be redrawn, not be copied from the original publications.

Author Response

We thank the reviewer for his comments.

"Fig" has been converted into "Figure" in the whole text, to improve consistency.

Concerning the figures: as much as we could, we redrawn the chemical structures of tweezers, but we copied the schemes illustrating their concept, mechanism or pharmaceutical application. In this case, we added "adapted from ref" in the legend. All the authorizations have been asked and obtained from publisher.

If this is not convenient, could you precise which figures needs to be redrawn ?

Reviewer 2 Report

In this manuscript authors review the reports on molecular tweezers, clefts, clips, and switches that are applicable for pharmaceutical molecular recognition, sensing, enzyme inhibition, molecular aggregation, etc. They introduced the brief history of these molecular apparatuses, and they noted their bindings to DNAs, proteins, sugars and lipids while showing recent publications. The present manuscript is adequately prepared and easy to read. Therefore, the paper will be suitable for publication.

I found some mistakes in writing, and I would like to ask authors revise the manuscript before publication.

Page 16, Figure 8 "Cation detection by fluorescence":

 Ref. [84] may be more suitable for compound 31.

Page 16, Line 2 from the bottom:

 (6.5-7) >> (pH 6.5-7)

Page 19, Figure 9:

 compound 36: "2+" is put nearby the oxo (O=) group. This may not be correct.

 compound 37: <sup>2+</sup>Zn >> Zn<sup>2+</sup>

Reference No.83:

  Chemistry >> Chem. Eur. J.

Author Response

We thank the reviewer for his comments. We have double checked the text and corrected the typos as much as we could.

Ref. [85] (former ref 84) has been changed for compound 31 in Figure 8.

Page 16, Line 2 from the bottom:

 (6.5-7) >> (pH 6.5-7)

 Thanks for noticing. We have corrected the text as suggested.

Page 19, Figure 9: We have corrected the 2+ to be close to the Calcium atom.

We could not change the 2+ of Zn on Figure 9

Reference No.84 has been corrected for Chem. Eur. J.

Reviewer 3 Report

review attached as a PDF file

Author Response

See attached document with detailed answers
